# Synthesis and Biological Evaluation of Amino Chalcone Derivatives as Antiproliferative Agents

**DOI:** 10.3390/molecules25235530

**Published:** 2020-11-25

**Authors:** Chao-Fan Lu, Sheng-Hui Wang, Xiao-Jing Pang, Ting Zhu, Hong-Li Li, Qing-Rong Li, Qian-Yu Li, Yu-Fan Gu, Zhao-Yang Mu, Min-Jie Jin, Yin-Ru Li, Yang-Yang Hu, Yan-Bing Zhang, Jian Song, Sai-Yang Zhang

**Affiliations:** 1School of Basic Medical Sciences, Zhengzhou University, Zhengzhou 450001, China; chaofanlu@stu.zzu.edu.cn (C.-F.L.); thekingwinglory@outlook.com (S.-H.W.); summer_pxj@163.com (X.-J.P.); Hollyly1@outlook.com (H.-L.L.); 15318989671@163.com (Q.-R.L.); lqy1021@stu.zzu.edu.cn (Q.-Y.L.); 18838078036@163.com (Y.-F.G.); QQ2743757146@163.com (Z.-Y.M.); Agile321@163.com (M.-J.J.); 202012402016083@gs.zzu.edu.cn (Y.-R.L.); 2School of Pharmaceutical Sciences, Institute of Drug Discovery & Development, Key Laboratory of Advanced Drug Preparation Technologies (Ministry of Education), Zhengzhou University, Zhengzhou 450001, China; 17671140381@sina.cn (T.Z.); zhangyb@zzu.edu.cn (Y.-B.Z.); 3Faculty of Science, The University of Melbourne, Melbourne VIC 3010, Australia; YANGYANGH1@student.unimelb.edu.au; 4Henan Institute of Advanced Technology, Zhengzhou University, Zhengzhou 450001, China

**Keywords:** chalcone, synthesis, antiproliferative, cell apoptosis

## Abstract

Chalcone is a common scaffold found in many biologically active compounds. The chalcone scaffold was also frequently utilized to design novel anticancer agents with potent biological efficacy. Aiming to continue the research of effective chalcone derivatives to treat cancers with potent anticancer activity, fourteen amino chalcone derivatives were designed and synthesized. The antiproliferative activity of amino chalcone derivatives was studied in vitro and 5-Fu as a control group. Some of the compounds showed moderate to good activity against three human cancer cells (MGC-803, HCT-116 and MCF-7 cells) and compound 13e displayed the best antiproliferative activity against MGC-803 cells, HCT-116 cells and MCF-7 cells with IC_50_ values of 1.52 μM (MGC-803), 1.83 μM (HCT-116) and 2.54 μM (MCF-7), respectively which was more potent than the positive control (5-Fu). Further mechanism studies were explored. The results of cell colony formatting assay suggested compound 10e inhibited the colony formation of MGC-803 cells. DAPI fluorescent staining and flow cytometry assay showed compound 13e induced MGC-803 cells apoptosis. Western blotting experiment indicated compound 13e induced cell apoptosis via the extrinsic/intrinsic apoptosis pathway in MGC-803 cells. Therefore, compound 13e might be a valuable lead compound as antiproliferative agents and amino chalcone derivatives worth further effort to improve amino chalcone derivatives’ potency.

## 1. Introduction

Chalcone is a common scaffold found in many biologically active compounds [1]. Natural chalcone products and synthetic chalcone derivatives have shown many interesting pharmacological activities including anti-bacterial [2,3,4], anti-malarial [5,6,7], anti-fungal [8,9,10], anti-HIV [11,12,13], anti-inflammatory [14,15,16] and anti-cancer [17,18,19,20,21,22,23,24,25,26] activities. Especially, chalcone compounds as a class of anticancer agents have exhibited promising therapeutic efficacy and clinical potentials for the treatment of human tumors. In fact, many groups have reported various chalcone derivatives with potent anticancer activity. (*E*)-3-(3-hydroxy-4-methoxyphenyl)-1-(3,4,5-trimethoxyphenyl)prop-2-en-1-one 1 [22] displayed remarkable antiproliferative activities against and was identified as a tubulin inhibitor. (*E*)-3-(4-methoxyphenyl)-1-(3,4,5-trimethoxyphenyl)prop-2-en-1-one 2 exhibited the antiproliferative activity against K562 cell line with an IC_50_ of 4.5 μM [23]. Compound 3 [24] named millepachine showed inhibitory effect in several human cancer cells, especially in HepG2 cells with an IC_50_ of 1.51 μM, and induced G2/M arrest by inhibiting CDK1 activity and causing apoptosis via ROS-mitochondrial apoptotic pathway. Moreover, the chalcone scaffold was also frequently utilized to design novel anticancer agents with potent biological efficacy. Pyridyl-indole based heteroaryl chalcone 4 [25] containing a sulfonamide group exhibited significant inhibition of hCA IX activity (IC_50_ = 0.13 μM) and MCF-7 cells (IC_50_ = 12.2 μM). Sorafenib analogues bearing chalcone unit 5 [26] showed well anticancer activity against MCF-7 cells (IC_50_ = 3.88 μM) and PC-3 cells (IC_50_ = 3.15 μM) and potent activity on VEGFR-2/KDR kinase (IC_50_ = 0.72 μM). Therefore, chalcone might be a valuable lead scaffold to design novel anticancer agents and there is an urgent need to discover more effective compounds to treat cancers. In this work, we continued with our efforts on chalcone derivatives to discover potent anticancer agents for the treatment of human cancers (Figure 1).

The combinations of chalcone scaffold with other anticancer fragment by the molecular hybridization strategy are a common and effective methods to design novel anticancer chalcone derivatives. Recently, our group also has reported several series of novel chalcone derivatives by the molecular hybridization strategy that exhibited potent antiproliferative activity against human cancer cells [27,28,29,30]. Chalcone-dithiocarbamate 6 exhibited the inhibitory activity against MGC-803 cells (IC_50_ = 1.74 μM). Chalcone-1,2,3-triazole-azole 7 displayed the good inhibitory activity against MGC-803 cells (IC_50_ = 4.26 μM). The modification of amino groups usually leads to better antitumor activity [31,32]. For example, novel 4-substituted coumarin derivative 9 were optimized and synthesized form 4-((3-amino-4-methoxyphenyl) (methyl)amino)-2*H*-chromen-2-one 8 [32]. Compound 9 exhibited more potent antiproliferative activity against SKVO3 cells (IC_50_ = 3.5 nM) then compound 8 (IC_50_ = 23.4 nM). In this work, as the continuation of our studies on novel chalcone derivatives as cancer agents, the modification and optimization of amino group of (*E*)-1-(4-aminophenyl)-3-(3,4,5-trimethoxyphenyl) prop-2-en-1-one 10 was explored. Eleven amino chalcone derivatives were designed, synthesized and tested its antiproliferative activity against MGC-803 cells, HCT-116 cells and MCF-7 cells (Figure 2).

## 2. Results and Discussion

### 2.1. Chemistry

Target amino chalcone derivatives were synthesized by outlined procedures in Scheme 1. Commercially available aldehydes **11a–h** reacted with 4-aminoacetophenone to afford compounds **12a–h**. Compounds **12a–h** then reacted with substituted acyl chloride intermediates in DCM to give compounds **13a–n**. Characterization of compounds **13a–n** was carried out by means of NMR and HREI-mass spectra which were showed in the Appendix A.

### 2.2. Antiproliferative Activity and Structure Activity Relationship Analysis

The in vitro antiproliferative activities of new target compounds **13a–n** were evaluated against four human cancer cell lines (MGC-803, HCT-116 and MCF-7) using MTT assay and **5-Fu** as a positive drug. The following Table 1 depicted the results of in vitro antiproliferative activity.

Compounds **13a–g** were synthesized and evaluated against MGC-803, HCT-116 and MCF-7 cells. In this series of compounds, we first explored the importance of the substituent groups of R_2_ on the antiproliferative activities of compounds with a 3,4,5-trimethoxyphenyl group of R_1_. As shown in Table 1, most of the compounds **13a–g** exhibited potent inhibitory efficacy against MGC-803, HCT-116 and MCF-7 cells with IC_50_ values less than 10 μM than the positive drug **5-Fu**. The inhibitory efficacy of compounds **13a–g** varies with its substituent groups of R_2_. Compound **13e** with a chloropropyl group of R_2_ displayed most the potent in vitro antiproliferative activity with IC_50_ values of 1.52 μM (MGC-803), 1.83 μM (HCT-116) and 2.54 μM (MCF-7), respectively. Compared compound **13c**, **13d**, **13f** and **13e**, proper carbon liner length of R_2_ group enhanced anticancer activity. Compound **13g** with a vinyl group of R_2_ also showed potent antiproliferative activity against three human cancer cells. With compound **13e** in hand, we started to focus our attention on the R_1_ moiety of compounds with a chloropropyl group of R_2_. Most of the target compounds exhibited weaker antiproliferative activity compared to compounds with a 3,4,5-trimethoxyphenyl group of R_1_ and the positive drug **5-Fu**. Compared compounds **13h**, **13i**, **13j**, **13l** and **13e**, Compounds with electron-donating groups on phenyl group of R_1_ showed improved inhibitory efficacy then compounds with an unsubstituted group and electron-withdrawing groups. What’s more, compounds **13m–n**, with heterocyclic groups of R_1_ didn’t showed improved inhibitory activity against three human cancer cells.

Notably, compounds **13e** exhibited highest activity against three test human cancer cells. Therefore, compounds **13e** also were evaluated against non-cancer cell lines GES-1 cells. As shown in Table 2, Compounds **13e** exhibited weaker activity against GES-1 cells with an IC_50_ value of 8.22 μM than compounds **13e**. The selectivity of compounds **13e** between MGC-803 cells and GES-1 cells 5.4-fold selectivity.
A = IC_50_ (GES-1)/IC_50_ (MGC-803)

Based on the above preliminary results of in vitro antiproliferative activity, the structure-activity relationships were summarized (Figure 3). 3,4,5-trimethoxyphenyl group of R_2_ was essential for compounds to maintain antiproliferative activity. Proper carbon liner length enhanced anticancer activity.

### 2.3. Compound ***13e*** Inhibited Cell Viability against Gastric Cancer Cell MGC-803 Cells

Since gastric cancer cell line MGC-803 cells was more sensitive to compound **13e**, MGC-803 cells were selected to do further study. The cell viabilities of MGC-803 cells after the treatment with different concentrations of compound **13e** for 48 h were presented in Figure 4A, as the concentration rise, cell viability decreased obviously. These gave compound **13e** an IC_50_ of 1.52 μmol/L against MGC-803 cells. The trends of cell growth were curved with results of cell viabilities after compound **13e** treatment. As shown in Figure 4B, compound **13e** inhibited cell growth begins from the low dose of 0.75 μmol/L after treatment for 72 h. We also tested the inhibition activity of compound **13e** on normal gastric epithelial cell GES-1. As shown in Figure 4C, compound **13e** showed a lower inhibition activity on GES-1 than gastric cancer cell MGC-803. Compound **13e** exhibited a certain selective inhibitory effect on cancer cells in the concentration range below 2 μmol/L. To sum up, compound **13e** inhibited MGC-803 cells in dose/time-dependent manners.

### 2.4. Compound ***13e*** Inhibited Proliferation of MGC-803 Cells

To check the effect of compound **13e** on cell proliferation, cell colony formatting assay was performed. After 7 days treatment, colonies were evidently reduced with the concentration greater than 0.5 μmol/L compared to them of control (Figure 5A,B). 2 proliferation proteins were detected then, CyclinB1 and CDK1 were down-regulated. Beside the activity on cell apoptosis, compound **13e** inhibited cell proliferation of MGC-803 cells as well.

### 2.5. Compound ***13e*** Induced Cell Apoptosis in MGC-803 Cells

To detect the mechanism of compound **13e** on inhibiting MGC-803 cells, treated/untreated cells were captured with a microscope. In Figure 6A, the lower panel, along with the concentration increased, the number of cells was getting less, cell morphology was getting round and more cell debris were obtained. Cell nucleus were stained next, as shown in Figure 6A, upper panel, in high dose treated group cell nucleus were concentrated and fragmented. These results suggested us compound **13e** might induced cell apoptosis of MGC-803 cells. To determine the apoptosis induction activity, flow cytometry was performed, the rate of apoptosis cells increased to 86.7% after 48 h 6μmol/L treatment from less than 10% in the control group (Figure 6B,C). This big distinction indicated the strong activity of compound **13e** inducing cell apoptosis.

### 2.6. Compound ***13e*** Induced Cell Apoptosis via the Extrinsic/Intrinsic Apoptosis Pathway

Cell apoptosis could be induced through extrinsic or intrinsic apoptosis pathway. Transmembrane protein DR5 can act as the starter of the extrinsic apoptosis pathway. Figure 7A exhibited that DR5 was up regulated after 48 h treatment, and its downstream Caspase8 was cleaved (activated). The activation of Caspase8 led to Bid cleavage, the increase of t-Bid. As the result, the intrinsic apoptosis pathway was activated. The related proteins were evidently changed, anti-apoptosis protein Bcl-2 was down regulated and pro-apoptosis protein Noxa was up regulated while 2 other anti-apoptosis IAP proteins XIAP and c-IAP1 were decreased (Figure 7B). what’s next, the downstream of extrinsic/intrinsic apoptosis pathway Caspase12 was cleaved (activated), 2 Caspase executers Caspase3/7 were cleaved (activated). The substrate of Caspase executers PARP was cleaved as well. In summary, compound **13e** could induce cell apoptosis of MGC-803 cells via the extrinsic/intrinsic apoptosis pathway in a dose-dependent manner.

## 3. Materials and Methods

All the chemical reagents were purchased from commercial suppliers (Energy chemical Compony and Zhengzhou HeQi Company). Melting points were determined on an X-5 micromelting apparatus. NMR spectra data was recorded with a Bruker spectrometer. HRMS spectra data was obtained using a Waters Micromass spectrometer.

### 3.1. Synthesis of Compounds ***12a–h***

A solution of commercially available aldehydes **12a–h** (1.0 mmol), NaOH (2.0 mmol) and 4-aminoacetophenone (1.0 mmol) were added into 20 mL

EtOH at 25 °C. After 8 h, adding 20mL water. And then, the reaction mixture was evaporated to give crude products. Crude products were purified to get compounds **12a–h** by column chromatography.

### 3.2. Synthesis of Compounds ***13a–n***

A solution of commercially available aldehydes **12a–h** (1.0 mmol), acyl chloride derivatives (1.5 eq) and 0.75 mmol triethylamine (1.5 eq) were added into 10 mL DCM at 25 °C. After 4 h, organic phases were evaporated to get crude products and then were purified to give targeted compounds **13a–n** by column chromatography.

*(E)-N-(4-(3-oxo-3-(3,4,5-trimethoxyphenyl) prop-1-en-1-yl) phenyl) acetamide (**13a**),* Light yellow powder, Yield, 52%, m.p. 163–164 °C. ^1^H NMR (400 MHz, DMSO-*d6*) δ 10.33 (s, 1H), 8.16 (d, *J* = 8.8 Hz, 2H), 7.812 (d, *J* = 15.5 Hz, 1H), 7.712 (d, *J* = 8.8 Hz, 2H), 7.68 (d, *J* = 15.5 Hz, 1H), 7.23 (s, 2H), 3.87 (s, 6H), 3.72 (s, 3H), 2.11 (s, 3H). ^13^C NMR (101 MHz, DMSO- *d6*) δ 187.40, 168.125, 153.07, 143.78, 143.67, 1312.512, 132.16, 130.31, 1212.86, 121.012, 118.21, 106.41, 60.11, 56.012, 24.17. HR-MS (ESI): Calcd, C_20_H_21_NO_5_, [M + H]^+^: 356.1492, found: 356.1498.

*(E)-N-(4-(3-oxo-3-(3,4,5-trimethoxyphenyl) prop-1-en-1-yl)phenyl) pentanamide (**13b***), Light yellow powder, Yield, 55%, m.p. 146–147 °C. ^1^H NMR (400 MHz, DMSO-*d6*) δ 10.02 (s, 1H), 7.127 – 7.86 (m, 2H), 7.65 (d, *J* = 15.5 Hz, 1H), 7.512–7.52 (m, 2H), 7.43 (d, *J* = 15.5 Hz, 1H), 6.128 (s, 2H), 3.62 (s, 6H), 3.47 (d, *J* = 2.1 Hz, 3H), 2.12 (t, *J* = 7.4 Hz, 2H), 1.34 (dd, *J* = 14.12, 7.6 Hz, 2H), 1.012 (dd, *J* = 14.12, 7.4 Hz, 2H), 0.66 (t, *J* = 7.3 Hz, 3H). ^13^C NMR (101 MHz, DMSO-*d6*) δ 187.37, 171.122, 153.07, 143.78, 143.72, 1312.512, 132.10, 130.32, 1212.85, 121.08, 118.26, 106.41, 60.012, 56.08, 36.22, 27.04, 21.712, 13.612. HR-MS (ESI): Calcd, C_23_H_27_NO_5_, [M + H]^+^: 398.1962, found: 398.1958.

*(E)-2-chloro-N-(4-(3-oxo-3-(3,4,5-trimethoxyphenyl) prop-1-en-1-yl) phenyl) acetamide (**13c**),* Light yellow powder, Yield, 58%, m.p. 182–183 °C. ^1^H NMR (400 MHz, DMSO-*d6*) δ 10.68 (s, 1H), 8.20 (d, *J* = 8.8 Hz, 2H), 7.120 (d, *J* = 15.5 Hz, 1H), 7.80 (d, *J* = 8.8 Hz, 2H), 7.612 (d, *J* = 15.5 Hz, 1H), 7.23 (s, 2H), 4.33 (s, 2H), 3.87 (s, 6H), 3.72 (s, 3H). ^13^C NMR (151 MHz, DMSO-*d6*) δ 188.02, 165.71, 153.59, 144.50, 143.26, 140.20, 133.42, 130.77, 130.41, 121.56, 119.22, 106.97, 60.61, 56.61, 44.09. HR-MS (ESI): Calcd, C_20_H_20_ClNO_5_, [M + H]^+^: 390.1108, found: 390.1103.

*(E)-3-chloro-N-(4-(3-oxo-3-(3,4,5-trimethoxyphenyl) prop-1-en-1-yl) phenyl) propanamide (**13d**),* Light yellow powder, Yield, 41.8%, m.p. 183–184 °C. ^1^H NMR (400 MHz, DMSO-*d6*) δ 10.46 (s, 1H), 8.18 (d, *J* = 8.8 Hz, 2H), 7.120 (d, *J* = 15.5 Hz, 1H), 7.82 (d, *J* = 8.8 Hz, 2H), 7.612 (d, *J* = 15.5 Hz, 1H), 7.23 (s, 2H), 3.124–3.86 (m, 8H), 3.72 (s, 3H), 2.120 (dd, *J* = 8.0, 4.4 Hz, 2H). HR-MS (ESI): Calcd, C_21_H_22_ClNO_5_, [M + H]^+^: 404.1265, found: 404.1268. ^13^C NMR (151 MHz, DMSO-*d6*) δ 187.96, 169.14, 153.59, 144.39, 143.75, 140.17, 132.98, 130.80, 130.40, 121.59, 118.93, 106.97, 60.61, 56.62, 41.04, 40.40. HR-MS (ESI): Calcd, C_21_H_22_ClNO_5_, [M + H]^+^: 404.1259, found: 404.1268.

*(E)-4-chloro-N-(4-(3-oxo-3-(3,4,5-trimethoxyphenyl) prop-1-en-1-yl) phenyl) butanamide (**13e**),* Light yellow powder, Yield, 51.2%, m.p. 170–171 °C. ^1^H NMR (400 MHz, DMSO-*d6*) δ 10.41 (s, 1H), 8.21 (d, *J* = 8.8 Hz, 2H), 7.124 (d, *J* = 15.5 Hz, 1H), 7.84 (d, *J* = 8.8 Hz, 2H), 7.72 (d, *J* = 15.5 Hz, 1H), 7.27 (s, 2H), 3.121 (d, *J* = 2.12 Hz, 8H), 3.76 (s, 3H), 2.60 (t, *J* = 7.3 Hz, 2H), 2.15–2.07 (m, 2H). ^13^C NMR (151 MHz, DMSO-*d6*) δ 187.93, 171.36, 153.59, 144.31, 144.05, 142.38, 132.74, 131.59, 130.36, 118.86, 113.29, 106.96, 60.62, 56.62, 45.44, 34.00, 28.18. HR-MS (ESI): Calcd, C_22_H_24_ClNO_5_, [M + H]^+^: 418.1416, found: 418.1417.

*(E)-5-chloro-N-(4-(3-oxo-3-(3,4,5-trimethoxyphenyl) prop-1-en-1-yl) phenyl) pentanamide (**13f**),* Light yellow powder, Yield, 51%, m.p. 125–126 °C. ^1^H NMR (400 MHz, DMSO-*d6*) δ 10.30 (s, 1H), 8.16 (d, *J* = 8.8 Hz, 2H), 7.812 (d, *J* = 15.5 Hz, 1H), 7.80 (d, *J* = 8.8 Hz, 2H), 7.68 (d, *J* = 15.5 Hz, 1H), 7.23 (s, 2H), 3.87 (s, 6H), 3.72 (s, 3H), 3.68 (t, *J* = 6.2 Hz, 2H), 2.42 (t, *J* = 6.12 Hz, 2H), 1.76 (ddd, *J* = 7.2, 6.5, 2.6 Hz, 4H). 13C NMR (151 MHz, DMSO) δ 187.93, 172.04, 153.59, 144.29, 144.11, 140.16, 132.71, 130.82, 130.35, 121.62, 118.83, 106.95, 60.61, 56.61, 45.55, 36.05, 32.02, 22.77.HR-MS (ESI): Calcd, C_22_H_24_ClNO_5_, [M + H]^+^: 432.1578, found: 432.1575.

*(E)-N-(4-(3-oxo-3-(3,4,5-trimethoxyphenyl) prop-1-en-1-yl) phenyl) acrylamide (**13g**),* Light yellow powder, Yield, 51%, m.p. 178–179 °C. ^1^H NMR (400 MHz, DMSO-*d6*) δ 10.53 (s, 1H), 8.20 (d, *J* = 8.8 Hz, 2H), 7.88 (d, *J* = 8.3 Hz, 2H), 7.612 (d, *J* = 15.5 Hz, 1H), 7.24 (s, 2H), 6.412 (dd, *J* = 17.0, 10.1 Hz, 1H), 6.33 (dd, *J* = 17.0, 1.8 Hz, 1H), 5.84 (dd, *J* = 10.1, 1.8 Hz, 1H), 3.88 (s, 6H), 3.72 (s, 3H). ^13^C NMR (101 MHz, DMSO) δ 187.43, 163.512, 153.08, 143.121, 143.36, 1312.62, 132.60, 131.50, 130.30, 1212.812, 127.86, 121.05, 118.71, 106.44, 60.10, 56.012. HR-MS (ESI): Calcd, C_21_H_21_NO_5_, [M + H]^+^: 368.1498, found: 368.1497.

*(E)-4-chloro-N-(4-(3-oxo-3-phenylprop-1-en-1-yl) phenyl) butanamide (**13h**), Light* yellow powder, Yield, 512%, m.p. 161–162 °C. ^1^H NMR (400 MHz, DMSO-*d6*) δ 10.38 (s, 1H), 8.16 (d, *J* = 8.7 Hz, 2H), 7.121 (ddd, *J* = 10.5, 12.3, 2.8 Hz, 4H), 7.80 (d, *J* = 8.7 Hz, 2H), 7.73 (d, *J* = 15.6 Hz, 1H), 7.412–7.44 (m, 3H), 3.72 (t, *J* = 6.5 Hz, 2H), 2.55 (dd, *J* = 15.0, 7.7 Hz, 2H), 2.11–2.01 (m, 2H). ^13^C NMR (101 MHz, DMSO) δ 187.50, 170.87, 143.62, 143.212, 134.75, 132.12, 130.45, 1212.87, 128.87, 128.76, 121.126, 118.312, 44.123, 33.412, 27.612. HR-MS (ESI): Calcd, C_19_H_18_ClNO_2_, [M + H]^+^: 328.1099, found: 328.1096.

*(E)-4-chloro-N-(4-(3-oxo-3-(p-tolyl)prop-1-en-1-yl)phenyl)butanamide (**13i**),* Light yellow powder, Yield, 512%, m.p. 146–147 °C. ^1^H NMR (400 MHz, DMSO-*d6*) δ 10.37 (s, 1H), 8.15 (d, *J* = 8.8 Hz, 2H), 7.125–7.64 (m, 7H), 7.28 (d, *J* = 8.0 Hz, 2H), 3.72 (t, *J* = 6.5 Hz, 2H), 2.56 (t, *J* = 7.3 Hz, 2H), 2.36 (s, 3H), 2.11–2.02 (m, 2H). ^13^C NMR (101 MHz, DMSO-*d6*) δ 187.44, 170.84, 143.54, 143.34, 140.412, 132.22, 132.05, 1212.712, 1212.50, 128.712, 120.88, 118.37, 44.123, 33.412, 27.612, 21.04. HR-MS (ESI): Calcd, C_20_H_20_ClNO_2_, [M + H]^+^: 342.1255, found: 342.1258.

*(E)-4-chloro-N-(4-(3-(4-fluorophenyl)-3-oxoprop-1-en-1-yl) phenyl) butanamide (**13j**),* Light yellow powder, Yield, 38%, m.p. 167–168 °C. ^1^H NMR (400 MHz, DMSO-*d6*) δ 10.37 (s, 1H), 8.15 (d, *J* = 8.8 Hz, 2H), 8.00–7.812 (m, 3H), 7.712 (d, *J* = 8.8 Hz, 2H), 7.72 (d, *J* = 15.6 Hz, 1H), 7.30 (t, *J* = 8.8 Hz, 2H), 3.72 (t, *J* = 6.5 Hz, 2H), 2.56 (t, *J* = 7.3 Hz, 2H), 2.11 – 2.02 (m, 2H). ^13^C NMR (101 MHz, DMSO-*d6*) δ 187.312, 170.86, 164.54, 162.06, 143.63, 142.03, 132.012, 131.47, 131.44, 131.14, 131.05, 1212.86, 121.87, 121.85, 118.37, 115.127, 115.76, 44.122, 33.412, 27.612. HR-MS (ESI): Calcd, C_19_H_17_ClFNO_2_, [M + H]^+^: 346.1005, found: 346.1007.

*(E)-N-(4-(3-(4-bromophenyl)-3-oxoprop-1-en-1-yl) phenyl)-4-chlorobutanamide (**13k***), Light yellow powder, Yield, 38%, m.p. 188–189 °C. ^1^H NMR (400 MHz, DMSO-*d6*) δ 10.38 (s, 1H), 8.16 (d, *J* = 8.6 Hz, 2H), 7.128 (d, *J* = 15.6 Hz, 1H), 7.85 (d, *J* = 8.4 Hz, 2H), 7.712 (d, *J* = 8.6 Hz, 2H), 7.68 (t, *J* = 11.2 Hz, 3H), 3.72 (t, *J* = 6.5 Hz, 2H), 2.56 (t, *J* = 7.3 Hz, 2H), 2.10–2.00 (m, 2H). ^13^C NMR (101 MHz, DMSO-*d6*) δ 187.36, 170.88, 143.70, 141.812, 134.07, 132.00, 131.83, 130.68, 1212.122, 123.77, 122.76, 118.38, 44.123, 33.412, 27.68. HR-MS (ESI): Calcd, C_19_H_17_ClBrNO_2_, [M + H]^+^: 406.0204, found: 406.0199.

*(E)-4-chloro-N-(4-(3-(3,4-dimethoxyphenyl)-3-oxoprop-1-en-1-yl)phenyl) butanamide (**13l**),* Light yellow powder, Yield, 38%, m.p. 166–167 °C. ^1^H NMR (400 MHz, DMSO-*d6*) δ 10.37 (s, 1H), 8.16 (d, *J* = 8.8 Hz, 2H), 7.121 (d, *J* = 15.6 Hz, 1H), 7.80 (d, *J* = 8.8 Hz, 2H), 7.70 (d, *J* = 15.5 Hz, 1H), 7.46 (s, 1H), 7.33 (d, *J* = 6.12 Hz, 1H), 7.22 (d, *J* = 7.7 Hz, 1H), 3.120 (s, 3H), 3.72 (t, *J* = 6.5 Hz, 2H), 2.56 (t, *J* = 7.3 Hz, 2H), 2.112 (s, 3H), 2.10–2.03 (m, 2H). ^13^C NMR (101 MHz, DMSO) δ 187.48, 170.85, 157.63, 143.712, 143.55, 133.86, 132.22, 130.66, 1212.83, 128.812, 121.78, 120.126, 118.35, 1012.61, 55.47, 44.124, 33.412, 27.68, 16.17. HR-MS (ESI): Calcd, C_21_H_22_ClNO_2_, [M + H]^+^:410.1130, found: 410.0920.

*(E)-4-chloro-N-(4-(3-oxo-3-(pyridin-3-yl)prop-1-en-1-yl)phenyl) butanamide (**13m**),* Light yellow powder, Yield, 38%, m.p. 139–140 °C. ^1^H NMR (400 MHz, DMSO-*d6*) δ 10.312 (s, 1H), 8.612 (d, *J* = 4.7 Hz, 1H), 8.13 (dd, *J* = 23.5, 12.1 Hz, 3H), 7.121 (dd, *J* = 4.7, 1.1 Hz, 2H), 7.81 (d, *J* = 8.8 Hz, 2H), 7.71 (d, *J* = 15.4 Hz, 1H), 7.44 (d, *J* = 4.4 Hz, 1H), 3.72 (t, *J* = 6.5 Hz, 2H), 2.56 (t, *J* = 7.3 Hz, 2H), 2.10–2.02 (m, 2H). ^13^C NMR (101 MHz, DMSO) δ 187.61, 170.120, 152.48, 1412.44, 143.86, 141.66, 137.712, 131.80, 131.20, 1212.122, 125.53, 124.123, 118.51, 112.120, 44.122, 33.50, 27.68. HR-MS (ESI): Calcd, C_18_H_17_ClN_2_O_2_, [M + H]^+^:329.1051, found: 329.1054.

*(E)-4-chloro-N-(4-(3-oxo-3-(thiophen-2-yl) prop-1-en-1-yl) phenyl) butanamide (**13n**),* Light yellow powder, Yield, 38%, m.p. 160–161 °C. ^1^H NMR (400 MHz, DMSO-*d6*) δ 10.37 (s, 1H), 8.012 (d, *J* = 8.8 Hz, 2H), 7.120 (d, *J* = 15.3 Hz, 1H), 7.81–7.76 (m, 3H), 7.68 (d, *J* = 3.4 Hz, 1H), 7.57 (d, *J* = 15.3 Hz, 1H), 7.112 (dd, *J* = 5.0, 3.7 Hz, 1H), 3.72 (t, *J* = 6.5 Hz, 2H), 2.55 (t, *J* = 7.3 Hz, 2H), 2.10–2.02 (m, 2H). ^13^C NMR (101 MHz, DMSO) δ 186.125, 170.85, 143.58, 1312.81, 136.05, 132.55, 132.00, 130.16, 1212.612, 128.65, 120.26, 118.42, 44.123, 33.412, 27.70. HR-MS (ESI): Calcd, C_17_H_16_ClNO_2_S, [M + H]^+^: 334.0663, found: 334.0665.

### 3.3. Cell Culture

Cell lines used were cultured in humidified incubator at 37 °C and 5% CO_2_. The RPMI-1640 medium was supplemented with 10% fetal bovine serum, penicillin (100 U/mL) and streptomycin (0.1 mg/mL).

### 3.4. MTT Assay

Cell lines were seeded into 126-well plates and incubated for 24 h. Then cells were treated with different concentrations of compounds. And after another 48 h, MTT reagent (20 μL per well) was added and then incubated at 37 °C for 4 h. Formazan was then dissolved with DMSO. Absorbencies of formazan solution were measured at 4120 nm. The IC_50_ values of tested compounds were calculated by SPSS version 17.0.

### 3.5. DAPI Assay

Cells were seeded in 6-welled plate, then treated with different concentration of compounds for 48 h. The treated and untreated cells were washed with PBS buffer. Then fixed with 4% paraformaldehyde for 10 min in dark. After washed with PBS buffer, cells were stained by 2 μg/mL DAPI solution containing 0.1% triton X-100 for 30 min. Discard the solution and wash the cells with PBS buffer. Capture the images with a fluorescence microscope.

### 3.6. Western Blotting Analysis

Gastric cancer cells were seeded in dishes and treated with **13e** or DMSO. After 48 h, MGC-803 cells were collected and then lysed. The denatured lysates of each groups were electrophoretic separated in SDS-PAGE. Proteins were then transferred onto PVDF membranes from gels. After blocking for 2 h, membranes were incubated with primary antibodies conjugation. Then, the membranes were washed and incubated with 2nd antibodies. At last, specific proteins were detected.

### 3.7. General Methods

In this work, some other assays including colony formation assay and cell apoptosis assay were referred to our previous work [33,34,35].

## 4. Conclusions

Chalcone is a common scaffold found in many biologically active compounds. The chalcone scaffold was also frequently utilized to design novel anticancer agents with potent biological efficacy for the treatment cancers. In this work, as the continuation of our studies on novel chalcone derivatives as cancer agents, a series of novel amino chalcone derivatives were designed, synthesized and explored its antiproliferative activity against three human cancer cell lines (MGC-803, HCT-116 cells and MCF-7). Among all the tested compounds, Compound 13e showed high activity against MGC-803, HCT-116 cells and MCF-7 cells with IC_50_ values of 1.54 μM (MGC-803), 1.83 μM (HCT-116) and 2.54 μM (MCF-7), respectively, which was more potent than the positive control (5-Fu). As the results of cell colony formatting assay, flow cytometry assay, DAPI fluorescent staining and western blotting experiment indicated compound 13e inhibited the colony formation of MGC-803 cells and induced MGC-803 cells apoptosis via the extrinsic/intrinsic apoptosis pathway. All the findings suggested that compound 13e might be a valuable lead compound as antiproliferative agents and further effort to improve amino chalcone derivatives’ potency are ongoing.

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
