# Peer review of "Synthesis and Biological Evaluation of Amino Chalcone Derivatives as Antiproliferative Agents"

_molecules, 2020, doi:10.3390/molecules25235530_

Round 1
Reviewer 1 Report
The study from Lu and colaborares describes the synthesis and cytotoxic evaluation of new amino chalcones derivatives. Overall, the results are interesting to the readers of Molecules. However, major modifications are need before the publication of these research. Some suggestion for this manuscript are presented below:
Abstract
The abstract was too short and some details that could be interesting for the readers are not provided. Please make it more informative by answer these points:
- Please provide some background and/or motivations for this studies in the abstract;
- How many compounds did you evaluate is this study? Please indicate this in the abstract.
- What type of assays did you employ in these “Further mechanism studies”?
Introduction
- The introduction section is also too short and do not provide the major problem to be solved for these new chalcone derivatives: the need for effective compounds to treat cancer. The authors should present this problematic situation in first place.
- Please also note that the use of the terms “compound 1” or “compound 2” is confusing for the readers. At least for this reviewer, it is not clear if the “compound 1” exposed in the introduction is the same used in this study or is only the name given by the authors of the other study. Please clarify this point.
Results and Discussion - Please explain how to classify the cytotoxic action as moderate or good. What parameter did you use in this classification?
- The data are not discussed.
Methods
- The authors do not provide the methodology used in the assays using fluorescent probes.
- The cytotoxic evaluation should include non-cancer cell lines in order to show the specificity of the compound towards cancer cells.
Author Response
- The abstract was too short and some details that could be interesting for the readers are not provided. Please make it more informative by answer these points:
- Please provide some background and/or motivations for this study in the abstract;
- How many compounds did you evaluate is this study? Please indicate this in the abstract.
- What type of assays did you employ in these “Further mechanism studies”?
Response: Thanks for your valuable comment. As you suggested, we have rewritten the abstract part to be more interesting for the readers. In the revised manuscript, we have added the description of some background for this study, total numbers of compounds and type of assays did in the revised manuscript.
- The introduction section is also too short and do not provide the major problem to be solved for these new chalcone derivatives: the need for effective compounds to treat cancer. The authors should present this problematic situation in first place.
Response: Thanks for your valuable comment. As you suggested, we have rewritten and added some sentence of introduction section to provide the major problem to be solved for these new chalcone derivatives: the need for effective compounds to treat cancer in the revised manuscript.
- Please also note that the use of the terms “compound 1” or “compound 2” is confusing for the readers. At least for this reviewer, it is not clear if the “compound 1” exposed in the introduction is the same used in this study or is only the name given by the authors of the other study. Please clarify this point.
Response: Thanks for your valuable comment. We are very sorry for our negligence. In general, compounds were named compound 1, compound 2, compound 3 and so on according to the order in which they appeared in the introduction section in many pharmaceutical chemistry papers. We also have changed the terms of compound 1, compound 2 in the revised manuscript.
- Please explain how to classify the cytotoxic action as moderate or good. What parameter did you use in this classification?
Response: Thanks for your valuable comment. In this work, we used a first line anticancer drug 5-Fu as a positive control drug, therefore, we described the moderate or good antitumor activity of the compounds based on the antitumor activity of 5-Fu. However, we have changed the related description about the cytotoxic action in the revised manuscript.
- The data are not discussed.
Response: Thanks for your valuable comment. We are very sorry for our negligence about the characterization of compounds. The NMR and MS analysis date have been provided in the revised supporting information.
- The authors do not provide the methodology used in the assays using fluorescent probes. The cytotoxic evaluation should include non-cancer cell lines in order to show the specificity of the compound towards cancer cells.
Response: Thanks for your valuable comment. We are very sorry for our negligence about the methodology used in the assays. We have provided the methodology used in the assays using fluorescent probes in the revised manuscript. Moreover, the cytotoxic evaluation of compound 13e against non-cancer cell lines GES-1 cells was explored in the revised manuscript.
Reviewer 2 Report
The paper describes the synthesis and biological evaluation of some chalcone derivatives as antiproliferative agents. The products described are new, however their characterisation is inadequate (see below). The presentation is quite poor, as the structures in the images have different sizes, the font size of the text is not consistent, there are gaps in the text etc.
Moreover, compounds 10a and 10c are reported in the literature therefore their characterisation data should be compared to the literature and the relevant references added.
I suggest that the paper is accepted after the following corrections are done.
Corrections:
- According to the name of compound 10a in the experimental R2 should be a CH3 not H so I assume this is wrong in scheme 1.
- The products investigated are not fully characterised. In particular, IR, MS and elemental analysis data are missing. Those are required in order to verify the biological data and need to be added in the experimental. Otherwise, without proof of purity the biological data cannot be trusted and the paper should be rejected.
- The references section needs editing. Some references have information missing such as titles or page ranges.
- No supporting information file is provided. This is required. Please include a supporting information file with all the NMR spectra.
Author Response
- The presentation is quite poor, as the structures in the images have different sizes, the font size of the text is not consistent, there are gaps in the text etc. Moreover, compounds 10a and 10c are reported in the literature therefore their characterisation data should be compared to the literature and the relevant references added.
Response: Thanks for your valuable comment. We are very sorry for our mistakes and negligence about this manuscript. We have corrected the mistakes such as the structures, the font size of the text and the gaps in the text in the revised manuscript. The characterization of compounds 13a and 13C have been provided in the revised supporting information file.
- According to the name of compound 10a in the experimental R2 should be a CH3 not H so I assume this is wrong in scheme 1.
Response: Thanks for your valuable comment. We are very sorry for our mistakes and we have corrected the mistakes in the revised manuscript.
- The products investigated are not fully characterized. In particular, IR, MS and elemental analysis data are missing. Those are required in order to verify the biological data and need to be added in the experimental. Otherwise, without proof of purity the biological data cannot be trusted and the paper should be rejected.
Response: Thanks for your valuable comment. We are very sorry for our negligence about the characterization of compounds. The NMR and MS analysis date have been provided in the revised supporting information.
- The references section needs editing. Some references have information missing such as titles or page ranges
Response: Thanks for your valuable comment. We are very sorry for our negligence about the references section. We have corrected the mistakes in the revised manuscript.
- No supporting information file is provided. This is required. Please include a supporting information file with all the NMR spectra.
Response: Thanks for your valuable comment. We are very sorry for our negligence about the supporting information. We have provided supporting information file.
Round 2
Reviewer 1 Report
The authors have improved the quality of manuscript presentation, following our suggestions. However, they need to improve the use of formal language throughout the manuscript. Few suggestions:
- Lines 15-16: I suggest to change for “Aiming to continue the research of effective chalcone derivatives to treat cancer with potent anticancer activity,...”
- Line 61: the expression “What’s more” is not appropriate in formal sentences.
- Please provide the IC50 of Compound 13e toward MGC-803. With this data you could calculate the selective index.
- In general, the authors should provide more information in the figure legends. For instance, They should provide the legend for Figure 5b.
Author Response
- Lines 15-16: I suggest to change for “Aiming to continue the research of effective chalcone derivatives to treat cancer with potent anticancer activity,...”
Response: Thanks for your valuable comment. As you suggested, we have changed for “Aiming to continue the research of effective chalcone derivatives to treat cancer with potent anticancer activity,...” in lines 15-16 in the revised manuscript.
- Line 61: the expression “What’s more” is not appropriate in formal sentences.
Response: Thanks for your valuable comment. As you suggested, we have rewritten this sentence in the revised manuscript.
- Please provide the IC50 of Compound 13e toward MGC-803. With this data you could calculate the selective index.
Response: Thanks for your valuable comment. As you suggested, we have provided the IC50 of compound 13e toward MGC-803 cells and non-cancer cell lines GES-1 cells in the revised manuscript.
- In general, the authors should provide more information in the figure legends. For instance, They should provide the legend for Figure 5b.
Response: Thanks for your valuable comment. As you suggested, we have provided more information in the figure legends in the revised manuscript.
Reviewer 2 Report
The manuscript has now been improved.
The authors have now provided a supporting information file however this needs to be improved. Please have one NMR spectrum per page and use landscape configuration so as to have it clearly visible. Each spectrum needs to have a title and some information about the spectrum. Moreover, please provided zoomed images for all regions where the peaks are not clearly visible. Also, all peaks need to be identified, not only the ones that belong to the compound.
Author Response
- The authors have now provided a supporting information file however this needs to be improved. Please have one NMR spectrum per page and use landscape configuration so as to have it clearly visible. Each spectrum needs to have a title and some information about the spectrum. Moreover, please provided zoomed images for all regions where the peaks are not clearly visible. Also, all peaks need to be identified, not only the ones that belong to the compound.
Response: Thanks for your valuable comment. As you suggested, we have corrected it in the revised supporting information file.